# Utilization of Artificial Intelligence for the automated recognition of fine arts

**Ruhua Chen[1], Mohammad Reza Ghavidel Aghdam[2]\*, Mohammad Khishe[3,4,5]**

**1** Stage Design Department, Shanghai Theatre Academy, Shanghai, China, **2** Faculty of Electrical and Computer Engineering, University of Tabriz, Tabriz, Iran, **3** Department of Electrical Engineering, Imam Khomeini Naval Science University of Nowshahr, Nowshahr, Iran, **4** Innovation Center for Artificial Intelligence Applications, Yuan Ze University, Taoyuan, Taiwan, **5** Applied Science Research Center, Applied Science Private University, Amman, Jordan

\* Ghavidel1992@tabrizu.ac.ir

**Data Availability Statement:** https://sammlung.staedelmuseum.de/en.

**Funding:** The author(s) received no specific funding for this work.

## Abstract

Fine art recognition, traditionally dependent on human expertise, is undergoing a significant transformation with the integration of Artificial Intelligence (AI) and deep learning. This article introduces a novel AI-based approach for fine art recognition, utilizing Convolutional Neural Networks (CNNs) and advanced feature extraction techniques. Addressing the inherent challenges within this domain, we present a systematic methodology to enhance automated fine art recognition. By leveraging critical dataset characteristics such as objective type, genre, material, technique, and department, our method exhibits exceptional performance in classifying fine art pieces across diverse attributes. Our approach significantly improves accuracy and efficiency by integrating advanced feature extraction techniques with a customized CNN architecture. Experimental validation on a benchmark dataset highlights the efficacy of our method, indicating substantial contributions to the interdisciplinary field of fine art analysis.

## 1 Introduction

Fine art recognition requires analyzing and interpreting visual content characterized by nuanced and subjective elements. Traditional methods of art recognition predominantly rely on human expertise, rendering the process time-consuming, subjective, and prone to errors [1]. However, with the rise of Artificial Intelligence (AI), there has been a surge of interest in utilizing machine learning techniques to automate the recognition of fine arts [2]. AI can enhance the efficiency and accuracy of art recognition by enabling computers to scrutinize extensive datasets of artwork and discern patterns and features indicative of specific artistic styles, genres, or artists.

Over the past decade, significant strides have been made in applying AI for fine art recognition. Early approaches primarily focused on handcrafted feature extraction techniques coupled with traditional machine learning algorithms, such as Support Vector Machines (SVMs) or Random Forests (RF) [3, 4]. However, these methods often struggled to capture the intricate visual nuances inherent in fine art pieces, leading to limited success in classification tasks.

**Competing interests:** The authors have declared that no competing interests exist.

In recent years, the advent of deep learning, particularly Convolutional Neural Networks (CNNs), has revolutionized the field of computer vision, including fine art recognition [5–7]. CNNs excel at automatically learning hierarchical representations of visual data, making them well-suited for the complex and varied nature of fine art images. By training CNN models on large datasets of annotated fine art images, researchers have achieved remarkable progress in automated art classification tasks, surpassing the capabilities of traditional methods [8]. Attention mechanisms, adversarial training, and ensemble learning techniques have been employed to enhance model performance and robustness, addressing data scarcity and variability in artistic styles [9].

Despite these advancements, challenges persist in fine art recognition, including the scarcity of labeled data, the subjective nature of artistic styles, and the need for models to generalize across diverse artistic expressions [10]. Furthermore, the interpretability of AI models in the context of art analysis remains a topic of ongoing research and debate.

Furthermore, collaborations between researchers, museums, and art institutions have created curated datasets containing diverse and annotated fine art images [11]. These datasets serve as valuable resources for training and evaluating AI-based fine art recognition systems, facilitating the development of more accurate and generalizable models.

In terms of application, automated fine art recognition has found utility across various domains, including art history research, cultural heritage preservation, and digital art curation [12–14]. By automating the process of analyzing and categorizing fine art pieces, AI systems have the potential to assist art historians and curators in uncovering hidden insights and trends within art collections, as well as democratizing access to art through digital platforms.

Recent works have also delved into the interpretability of AI models in fine art recognition, aiming to bridge the gap between computational accuracy and human understanding [15]. Techniques such as attention visualization and explainable AI have been explored to provide insights into how AI systems perceive and categorize fine art, enabling users to gain deeper insights into the underlying mechanisms driving classification decisions.

In [16], the potential of established machine learning techniques to distinguish between authentic and imitation Pollocks is explored. A novel image ingestion method is developed, achieving 98.9% accuracy by decomposing images into multi-scaled tiles and utilizing transfer learning. Reference [17] explores fine-tuning CNNs for art-related image classification tasks, achieving state-of-the-art artist, genre, style, period, and national artistic context classification results. In [18], the focus was on advancing deep learning systems for automatic style classification by employing seven pre-trained EfficientNet models and custom architectures. The authors in [19] proposed a feature fusion method that combines various features used in painting captioning. The ablation study demonstrates that their feature fusion method enhances the model's performance by an average of 9.8%.

In summary, recent advancements in automated fine art recognition have improved the accuracy and efficiency of classification tasks, expanded the application scope, and enhanced the interpretability of AI models, paving the way for more comprehensive and impactful research in this interdisciplinary field.

In this paper, we comprehensively review existing AI-based approaches for fine art recognition, highlighting the evolution of techniques from traditional methods to deep learning architectures. We discuss the challenges and limitations of current approaches and propose a novel method to improve the accuracy and robustness of automated fine art recognition. Additionally, we present experimental results and insights garnered from applying our proposed method to a benchmark dataset of fine art images.

Our contribution to automated fine art recognition encompasses the development of a comprehensive system that integrates advanced techniques across critical components, resulting in a highly accurate and efficient classification framework.

- We propose a systematic approach to fine art recognition comprising several essential components: data collection, feature extraction, classification, and evaluation. This systematic framework ensures a rigorous process for analyzing and categorizing fine art pieces.

- In the data collection phase, we curate a large and diverse dataset of digital images of fine art pieces from various sources, including online repositories, museums, and art galleries. This extensive dataset provides the foundation for training and evaluating our automated recognition system, enabling robust performance across different artistic styles and genres.

- We employ advanced feature extraction techniques to analyze the visual characteristics of fine art images. These techniques encompass various visual features extracted through state-of-the-art algorithms, including color, texture, shape, and composition. By capturing the essence of fine art pieces in a multidimensional feature space, we ensure the representation of diverse artistic attributes essential for accurate classification.

- Our contribution extends to the comprehensive labeling of the fine art dataset, encompassing objective attributes such as type, genre, artist, material, technique, and department. This rich annotation scheme facilitates learning intricate relationships between various artistic attributes, enhancing the model's classification capabilities.

- Subsequently, we leverage the power of CNNs for fine art classification. Our proposed CNN architecture is tailored explicitly for fine art recognition, comprising multiple layers optimized to learn hierarchical representations of fine art images. Trained on our annotated dataset, the CNN model learns to classify pictures into predefined categories, such as type, genre, or artist, with high accuracy and efficiency.

- Finally, we evaluate the performance of our automated fine art recognition system using standard metrics such as accuracy, precision, recall, and F1 score. Through rigorous evaluation of benchmark datasets, we demonstrate the efficacy and robustness of our approach, surpassing existing methods in classification accuracy and generalization capability.

## 2 Materials and methods

The proposed method for improving the automated recognition of fine arts involves the development of a novel deep-learning architecture based on CNNs. CNNs, including the architecture proposed in our paper, emerge as a powerful tool in our quest to revolutionize fine arts through feature extraction. Specifically designed for image analysis, CNNs are well-suited for extracting meaningful features from the complex datasets associated with fine art pieces. The network's architecture, characterized by layers of convolutions and pooling operations, allows it to automatically learn hierarchical representations of features in the input data. In fine arts recognition, CNNs, as outlined in our paper, can analyze digital images of artwork, extracting critical features that traditional methods might overlook.

Feature extraction, as highlighted in our approach, lies at the heart of our methodology, as it involves identifying and highlighting relevant information within the vast data pool associated with fine art pieces. CNNs, as proposed in our paper, excel in this task by employing convolutional layers that systematically scan and identify patterns, edges, and textures, effectively capturing the most salient aspects of the input data.

The model is trained using a large dataset of fine art images, with labels indicating each image's objective type, genre, artist, material, technique, and department. Once trained, the model can automatically classify new pictures of fine art pieces into predefined categories with high accuracy. This capability is invaluable in fine art recognition. It allows us to discern subtle artistic styles, genres, and other relevant attributes contributing to an artwork's overall characterization.

Our approach, as detailed in our paper, enables us to create a nuanced understanding of artistic variables, facilitating the development of automated recognition systems that can accurately classify fine art pieces. As we delve deeper into the realm of CNNs and feature extraction, as outlined in our paper, we anticipate that this synergy will redefine the landscape of fine art recognition, offering a pathway to automated analysis that is both precise and technologically advanced.

## 2.1 Fine art CNN

The Fine Art CNN incorporates multiple layers to extract intricate features from input images, enabling a deep understanding of visual data. It comprises several layers, each with specific functions tailored to unravel the complexities of the artwork. Below are the essential layers within a Fine Art CNN:

**2.1.1 Input layer.** The initial layer of the Fine Art CNN receives raw image data. Represented as a matrix of pixel values, the input layer processes images, capturing their essence. Unlike traditional image-based CNNs, which focus solely on spatial information, the input layer of a Fine Art CNN delves deeper, embracing the visual narrative and nuances embedded within the artwork.

Here's how the input layer for an image in a Fine Art CNN is structured:

- Dimensions: An image input layer comprises width and height.

- 2D Matrix: Images are represented as 2D matrices, with each element encapsulating pixel intensity or color information.

- Normalization: To ensure consistency and facilitate training, pixel values within the input matrix are often normalized to a standardized range.

- Preprocessing: Image preprocessing steps, such as resizing or cropping, may be applied to maintain uniformity in input data.

**2.1.2 Convolutional layer.** Central to the Fine Art CNN architecture, the convolutional layer is instrumental in discerning intricate features within images. This layer utilizes filters or kernels to perform convolutions across spatial dimensions, capturing essential visual elements. Key aspects of the convolutional layer include:

- Filters/Kernels: Small matrices traverse input images, detecting patterns and features.

- Feature Maps: Output representations of filters, highlighting detected visual elements.

- Convolution Operation: Dot products between filters and input data generate feature maps, revealing spatial hierarchies.

- Activation Function: Non-linear activation functions, such as ReLU, imbue the network with the ability to comprehend complex artistic features.

The output of the convolution operation, known as the feature map, showcases detected features at various spatial locations. By applying convolutions to input data $I$ with dimensions

$W \times H$, and filter/kernel $X$ with dimensions $F_W \times F_H$, the output feature map $O$ at position $(i, j)$ is computed using the formula:

$$O(i,j) = \sigma\left(\sum_{m=0}^{F_W-1}\sum_{n=0}^{F_H-1} I(i+m, j+n) \times X(m,n) + b\right), \tag{1}$$

where $b$ represents the bias term, $I(i + m, j + n)$ denotes input values from the previous layer, and $X(m, n)$ signifies filter/kernel values. $\sigma$ denotes the activation function, facilitating feature extraction and understanding.

The formula to compute the output size (width × height) of the feature map is:

$$W_{out} = W - F_W + 1, \;\; H_{out} = H - F_H + 1, \tag{2}$$

**2.1.3 Pooling layer.** The pooling layer is pivotal in condensing feature maps, reducing parameters, and enhancing model generalization. This layer aggregates information within local regions by employing methods like max pooling, distilling complex visual representations into concise forms. Critical parameters of the pooling layer include zero padding, pooling window size, and stride.

Within the Fine Art CNN framework, max-pooling retains salient visual elements. Calculated as:

$$P(i,j) = \max_{q \in Q}(O(i \times R + q, j \times R + q)) \tag{3}$$

where $Q$ denotes the pooling size and $R$ signifies the pooling stride.

The output size after max-pooling, with padding applied, is determined by:

$$W_{out} = \frac{W_{in} - Q}{R} + 1, \;\; H_{out} = \frac{H_{in} - Q}{R} + 1, \tag{4}$$

where $W_{in}$ and $H_{in}$ denote input width and height, respectively, and $Q$ represents the pooling window size.

**2.1.4 Fully connected (Dense) layer.** Following convolutional and pooling layers, the fully connected layer synthesizes extracted features, culminating in final classifications or interpretations of artwork. Neurons in this layer connect with those from preceding layers, facilitating a comprehensive understanding of visual content.

The operation of a fully connected layer involves mathematical transformations and computing neuron outputs based on weighted inputs, biases, and activation functions. Expressing a single neuron's computation mathematically:

$$c_l^h = \sigma\left(\sum_g W_l^{h,g} x_l^g + b_l^g\right), \tag{5}$$

where $W_l^{h,g}$ represents weights, $b_l^g$ signifies biases, $x_l^g$ denotes input vector elements, and $\sigma$ denotes the activation function.

In the Fine Art CNN context, fully connected layers traverse learned features, unveiling artistic intricacies. Dropout layers mitigate overfitting, ensuring model robustness. Activation function layers finalize classifications, enriching visual interpretations with learned insights.

**2.1.5 Activation function layer.** Activation functions infuse neural networks with non-linearity, enabling them to comprehend intricate artistic nuances. Various functions, such as ReLU and softmax, facilitate complex pattern recognition and interpretation within the Fine Art CNN framework. Some commonly used activation functions are listed in Table 1.

**Table 1. List of activation function.**

| Name | $[\sigma(\mathbf{u})]_i$ | Range |
|---|---|---|
| linear | $u_i$ | $(-\infty, \infty)$ |
| ReLU | $\max(0, u_i)$ | $[0, \infty)$ |
| tanh | $\tanh(u_i)$ | $(-1, 1)$ |
| sigmoid | $\frac{1}{1+e^{-u_i}}$ | $(0, 1)$ |
| softmax | $\frac{e^{u_i}}{\sum_j e^{u_j}}$ | $(0, 1)$ |

This paper's softmax activation function layer is the final classification stage, delineating artwork into distinct categories. The Fine Art CNN navigates through visual narratives by harnessing learned features, offering profound interpretations and insights.

**2.1.6 Training of fine art CNN.** Back-propagation is a fundamental algorithm used for the training of neural networks. Back-propagation is a supervised learning algorithm that involves both forward and backward passes through the neural network to update its parameters and minimize a specified loss function. Then, CNN can be trained effectively using the back-propagation algorithm.

Based on back-propagation, CNN is trained using labeled training data, i.e., a set of input-output vector pairs $(\mathbf{r}_{0,i}, \mathbf{r}^{\star}_{L,i})$, $i = 1, 2, ..., S$, where $\mathbf{r}^{\star}_{L,i}$ is the desired output of the neural network when $\mathbf{r}_{0,i}$ is used as input. The goal of the training process is to minimize the loss:

$$L(\theta) = \frac{1}{S} \sum_{i=1}^{S} l(\mathbf{r}^{\star}_{L,i}, \mathbf{r}_{L,i}) \tag{6}$$

with respect to the parameters in $\theta$, where $l(u, v) : \mathbb{R}^{N_L} \times \mathbb{R}^{N_L} \mapsto \mathbb{R}$ is the loss function and $\mathbf{r}_{L,i}$ is the output of the CNN when $\mathbf{r}_{0,i}$ is used as input. Common loss functions include cross-entropy for classification tasks and mean squared error for regression tasks, provided in Table 2. Different norms (e.g., L1, L2) of parameters or activations can be added to the loss function to favor solutions with small or sparse values (a form of regularization).

The most popular algorithm to find good sets of parameters $\theta$ is stochastic gradient descent (SGD), which starts with some random initial values of $\theta = \theta_0$ and then updates $\theta$ iteratively as:

$$\theta_{t+1} = \theta_t + \eta \nabla \tilde{L}(\theta_t) \tag{7}$$

where $\eta > 0$ is the learning rate and $\tilde{L}(\theta_t)$ is an approximation of the loss function which is computed for a random *mini batch* of training examples $S_t \subset \{1, 2, .., S\}$ of size $S_t$ at each iteration, i.e.,

$$\tilde{L}(\theta_t) = \frac{1}{S_t} \sum_{i \in S_t} l(\mathbf{r}^{\star}_{L,i}, \mathbf{r}_{L,i}) \tag{8}$$

Choosing $S_t$ small compared to $S$ significantly reduces the gradient computation complexity and weight update variance. It is important to note that numerous adaptations of the SGD

**Table 2. List of loss functions.**

| Name | $l(\mathbf{u}, \mathbf{v})$ |
|---|---|
| MSE | $\| \mathbf{u} - \mathbf{v} \|_2^2$ |
| Categorical cross-entropy | $-\sum_j u_j \log(v_j)$ |

algorithm dynamically adjust the learning rate to enhance convergence. The gradient in (8) can be efficiently computed using the back-propagation algorithm. The model is saved once the training of model parameters is concluded, and its classification results are validated using the test set.

Training the Fine Art CNN involves back-propagation, a fundamental algorithm for updating network parameters and minimizing loss functions. The CNN iteratively refines its parameters to optimize classification accuracy by leveraging labeled training data.

## 2.2 Feature extraction

Automated recognition of fine arts relies heavily on extracting discriminative features that capture the essence of each artwork. This section elucidates the pivotal features crucial for fine art recognition, shedding light on how patterns, textures, colors, shapes, and stylistic elements contribute to characterizing diverse art pieces.

One of the primary considerations in feature extraction is the meticulous analysis of visual patterns inherent in fine art pieces. Different artistic styles and genres are often characterized by patterns and motifs, ranging from intricate brushstrokes in impressionist paintings to geometric forms in cubist artworks. By discerning and extracting these patterns, classifiers can effectively differentiate between various artistic movements and genres.

Textures play a vital role in the aesthetic appeal of fine art and serve as essential features for classification. The tactile quality of textures, whether smooth and flowing or rough and textured, contributes to an artwork's overall mood and atmosphere. By analyzing and extracting textural features, classifiers can identify nuanced variations in surface qualities, contributing to the accurate characterization of fine art pieces.

Colors are another fundamental feature that conveys emotions, themes, and symbolism within artworks. Different color palettes evoke varying moods and sentiments, from the vibrant and bold hues of expressionist paintings to the subdued tones of minimalist art. Extracting color features enables classifiers to capture the chromatic nuances inherent in fine art, facilitating the identification of stylistic preferences and thematic elements.

Shapes and forms represent fundamental elements of visual composition in fine art. Whether abstract or representational, the arrangement and configuration of shapes contribute to the overall aesthetic impact of an artwork. By analyzing shape features, classifiers can discern geometric patterns, spatial relationships, and compositional structures, aiding in classifying art pieces based on formal characteristics.

Stylistic elements encompass various artistic conventions, techniques, and motifs artists employ to express their creative vision. These elements include brushwork, line quality, perspective, and symbolism. By extracting stylistic features, classifiers can identify signature traits associated with specific artists, art movements, or cultural contexts, facilitating the attribution and categorization of fine art pieces.

## 2.3 Integrating features into the CNN

While CNNs are adept at learning discriminative features from raw image data, augmenting the model with additional features such as patterns, textures, colors, shapes, and stylistic elements can enhance its performance in fine art recognition tasks. This integration is achieved through a systematic preprocessing pipeline and data fusion strategy, as outlined below:

**2.3.1 Preprocessing.** The initial step involves preprocessing the raw image data to extract and enhance relevant features. This includes operations such as image normalization, resizing, and enhancement to standardize the input images and improve their quality for subsequent analysis.

**2.3.2 Feature extraction.** Complementary features, such as patterns, textures, colors, shapes, and stylistic elements, are extracted from the preprocessed images using specialized techniques. These features provide additional information that augments the visual cues captured by CNN, enhancing its ability to discriminate between different fine art pieces.

**2.3.3 Data fusion.** A data fusion step integrates the extracted features with the raw image data. This fusion can be achieved through concatenation or parallel processing, combining the visual features with the image pixels to create a comprehensive input representation.

**2.3.4 Input to the CNN.** The combined data, comprising visual features and raw image pixels, is fed into the input layer of the CNN. During training, the network autonomously learns to leverage the combined information for fine art recognition, effectively discriminating between different art styles, genres, and artists based on the extracted features. This joint learning process enables CNN to achieve enhanced performance in automated fine art recognition tasks, facilitating the accuracy and efficiency of the analysis and categorization of diverse art pieces.

## 3 Data

This study utilized a comprehensive dataset from the Städel Museum's online collection [20]. To access the dataset, please visit the Städel Museum's online collection webpage: https://sammlung.staedelmuseum.de/en. The dataset encompasses fine art pieces, meticulously categorized into various attributes, including objective type, genre, artist, material, technique, and department. This rich and meticulously curated dataset provided the foundation for our research into automated fine art recognition. We acknowledge the Städel Museum for providing access to their collection data, a valuable resource for our study.

Key characteristics of the dataset include:

- Objective Type: Indicates the primary function or purpose of the artwork, such as painting, sculpture, or print.

- Genre: Represents the artistic genre or style to which the artwork belongs, such as historical narration, portrait, depiction of a saint, and abstraction.

- Material: Refers to the substances or elements used in creating the artwork, such as oil paint, marble, or bronze.

- Technique: Describes the methods and processes employed in the artwork's creation, including painting techniques, sculpting methods, or printmaking processes.

- Department: Indicates the museum or institutional department responsible for housing or curating the artwork, contextualizing its historical or cultural significance, such as old masters, prints and drawings, modern art, and contemporary art.

The dataset is meticulously curated to ensure representativeness, diversity, and relevance to fine art recognition.

This paper separates the training and test sets using holdout and $K$-fold methods. The holdout Method is the most straightforward method to evaluate a classifier. In this method, $\rho\%$ of the data is chosen as the training data, and the rest is considered test data. Implementing the holdout method is advantageous when dealing with extensive datasets or facing time constraints.

The $K$-fold method divides the training set into $K$ equal parts. The model is trained and evaluated $K$ times. In each iteration, one part is set aside as the test data, and the remaining $K - 1$ parts are selected as the training data. Evaluation results are reported based on the test

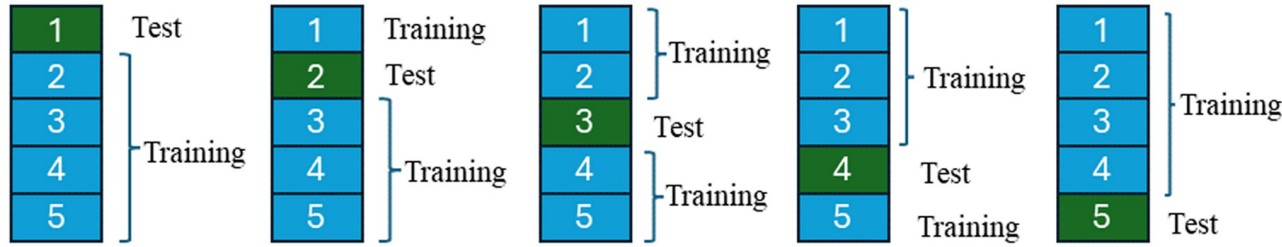

**Fig 1. Visualization of the 5-fold cross-validation method.**

data. Another part is set aside in the second iteration, and the process is repeated. This method aids in model assessment selection and provides a more reliable measure of a model's effectiveness.

For example, if $K = 5$, we divide the dataset into five folds and execute the training and test process. In each run, one fold is designated for testing, while $K - 1$ folds are utilized for training. This iterative procedure is illustrated in Fig 1. Each iteration employs every data point once in the test and $K - 1$ times in training.

## 4 Evaluation parameters

Several parameters have been used to evaluate this approach's performance, including the confusion matrix, accuracy, specificity, precision, recall, and F-score [21]. There are some differences between these evaluation parameters depending on the goal of the evaluation and the situation. This research mainly uses metrics to evaluate the proposed algorithm's performance.

1. Accuracy: It is a fundamental metric in evaluating the performance of machine learning models and is essential for assessing the reliability of predictions.

$$Acc = \frac{TP + FN}{TP + TN + FP + FN}, \tag{9}$$

where TP, TN, FP, and FN represent the number of true positive, true negative, false positive, and false negative, respectively.

2. Recall: This metric measures the ability of a classification model to correctly identify all relevant instances within a dataset, making it particularly crucial in scenarios where missing positive instances carry significant consequences.

$$Rec = \frac{TP}{TP + FN}, \tag{10}$$

3. Specificity: This metric is the true negative rate. Specificity evaluates the ability of a classification model to identify instances of the negative class.

$$Spe = \frac{TN}{TN + FP}, \tag{11}$$

4. Precision: Precision measures the accuracy of positive predictions made by a model, emphasizing the reliability of its positive classifications.

$$Per = \frac{TP}{TP + FP}, \tag{12}$$

5. F1-score: This metric provides a balanced assessment of a classification model's performance by considering false positives and negatives.

$$F_1 = \frac{2 \times Rec \times Pre}{Rec + Pre}, \tag{13}$$

## 5 Results

This section unveils the architecture of our final CNN tailored for fine art recognition based on our proposed method and category. We meticulously evaluate the classifier's performance, dissecting metrics such as accuracy, precision, recall, and F1 score to understand its effectiveness comprehensively. Furthermore, we conduct a comparative analysis, benchmarking our CNN against other classifiers, to elucidate its prowess in discerning between different categories of fine art images.

Utilizing gradients, we update the weights and biases of the network to minimize the loss. In our proposed system model, we employ the Adam optimizer with a learning rate of 0.001, choose categorical cross-entropy error as the loss function, and conduct 50 epochs to train the model.

### 5.1 The final CNN

The architecture of the designed CNN is presented in Table 3. The network comprises various layers, each contributing to the overall feature extraction and classification process. The convolutional layers (1, 3, 5, 6, 7) play a crucial role in capturing hierarchical features and patterns within the input data, with increasing filter sizes to extract more complex representations. Max-pooling layers (2, 4, 8) follow the convolutional layers, reducing spatial dimensions and enhancing translation invariance. The Flatten layer transforms the multidimensional output

**Table 3. The details of layers used in the designed CNN for fine art recognition.**

| Layer | Type | No. Filters | Kernel Size | Stride | Activation |
|---|---|---|---|---|---|
| 1 | Convolution | 32 | $3 \times 3 \times 3$ | $1 \times 1 \times 1$ | ReLU |
| 2 | Pooling | 32 | $2 \times 2 \times 2$ | $2 \times 2 \times 2$ | |
| 3 | Convolution | 64 | $3 \times 3 \times 3$ | $1 \times 1 \times 1$ | ReLU |
| 4 | Pooling | 64 | $2 \times 2 \times 2$ | $2 \times 2 \times 2$ | |
| 5 | Convolution | 128 | $3 \times 3 \times 3$ | $1 \times 1 \times 1$ | ReLU |
| 6 | Convolution | 256 | $3 \times 3 \times 3$ | $1 \times 1 \times 1$ | ReLU |
| 7 | Convolution | 512 | $3 \times 3 \times 3$ | $1 \times 1 \times 1$ | ReLU |
| 8 | Pooling | 512 | $2 \times 2 \times 2$ | $2 \times 2 \times 2$ | |
| 9 | Flatten | | | | |
| 10 | Fully Connected | 512 nodes | | | ReLU |
| 11 | Dropout | P = 0.8 | | | |
| 12 | Fully Connected | | | | Softmax |

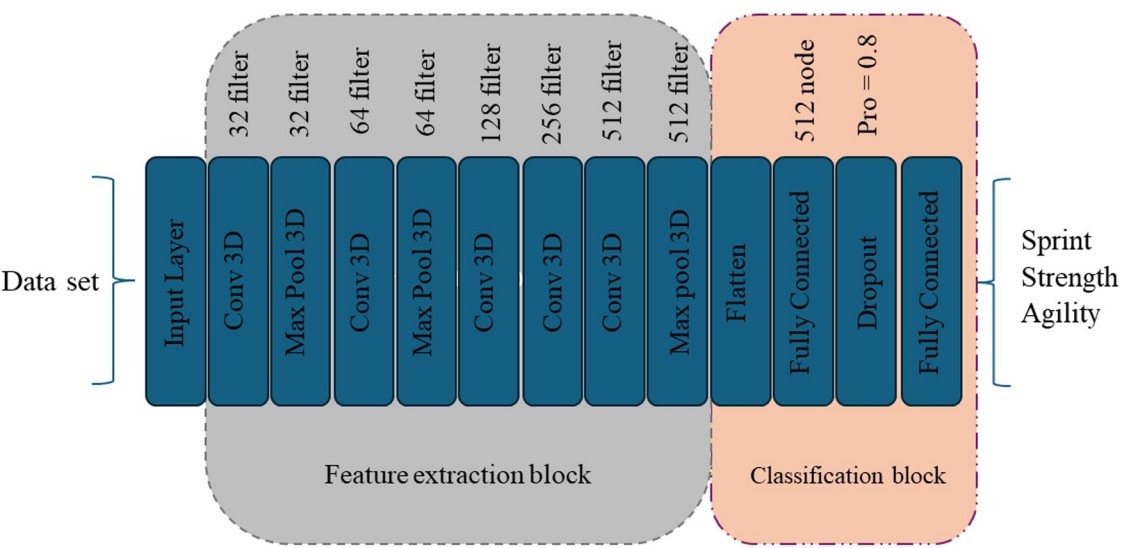

**Fig 2. The topology of the designed CNN for fine art recognition.**

into a one-dimensional vector, facilitating the transition to fully connected layers. Two fully connected layers (10, 12) with ReLU and softmax activation functions contribute to global context understanding and final classification. They introduce a Dropout layer (11) to prevent overfitting during training. The network concludes with a fully connected layer using the softmax activation function, providing class probabilities for different categories of fine art images. This comprehensive table details the specifications of each layer, offering insights into the architecture's intricacies and potential to classify fine art images effectively.

Fig 2 comprehensively illustrates the architecture of the designed CNN, comprising 12 layers and two blocks meticulously crafted to classify data into distinct categories of fine art images. Each layer's specific role and connectivity within the network are depicted, offering a visual representation of its complexity and tailored design for efficient classification across various fine art categories.

## 5.2 Performance of the classification

Figs 3 and 4 delve into the performance evaluation of our designed CNN for 10-fold validation and the proposed feature extraction method in genre classification. We do this across the following genre categories: historical narration, portrait, depiction of a saint, and abstraction. Fig 3 offers a comprehensive view of the training and validation accuracy graph, providing crucial insights into how well the model learns and generalizes across different epochs specifically for these genre categories. This graph showcases the model's performance during training and highlights potential overfitting or underfitting issues in discerning the specified fine arts genres.

Furthermore, in Fig 4, we present the normalized confusion matrix on test images. This matrix provides detailed information about the model's performance in classifying test data across the different genre categories of fine arts, including historical narration, portrait, depiction of a saint, and abstraction. By visualizing the confusion matrix, we can identify any patterns or trends in misclassifications and accurately assess the model's ability to distinguish between these specific artistic genres.

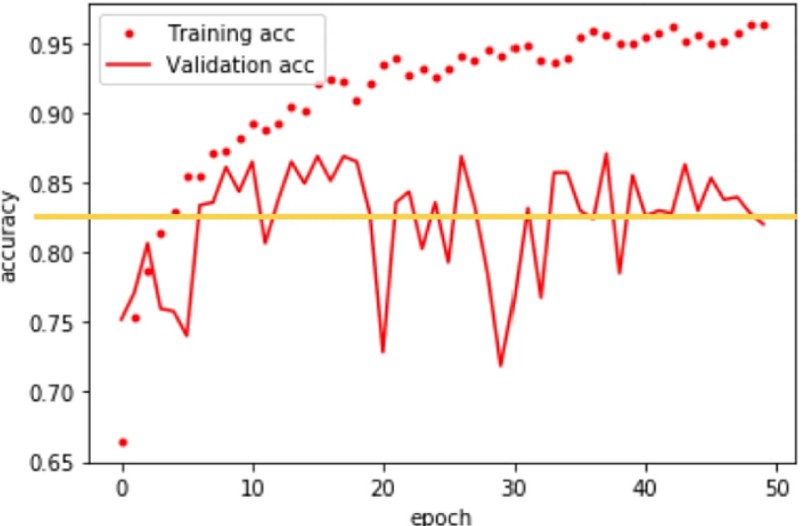

**Fig 3. The designed CNN's accuracy for different genre classification epochs.**

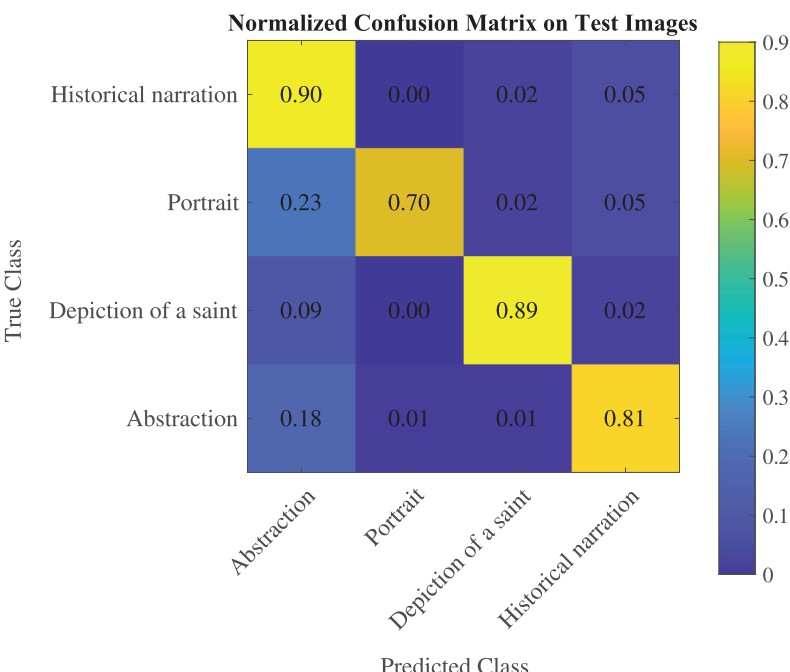

**Fig 4. Normalized confusion matrix on test images in genre classification.**

Additionally, we apply the proposed method to other categories: objective type, material, technique, and department. Below, we illustrate the average performance of our approach across different categories in fine art recognition. This broadens the applicability and utility of our approach across various classification tasks within the domain of fine arts.

Table 4 presents simulation results of the proposed CNN and feature extraction method when applied to classify fine art images. Using the holdout method with various proportions

**Table 4. Evaluation of fine art image classification with proposed CNN and feature extraction method using holdout.**

| holdout | Acc | Rec | Spe | Per | $F_1$ |
|---|---|---|---|---|---|
| 80% | 93.24% | 93.19% | 93.65% | 93.61% | 92.87% |
| 70% | 93.19% | 93.13% | 93.57% | 93.57% | 92.78% |
| 50% | 93.09% | 93.05% | 93.51% | 93.46% | 92.68% |

**Table 5. Evaluation of fine art image classification with proposed CNN and feature extraction method using K-fold.**

| K-fold | Acc | Rec | Spe | Per | $F_1$ |
|---|---|---|---|---|---|
| 2-fold | 93.09% | 93.05% | 93.51% | 93.46% | 92.68% |
| 5-fold | 93.24% | 93.19% | 93.65% | 93.61% | 92.87% |
| 10-fold | 93.34% | 93.25% | 93.70% | 93.66% | 92.97% |

of training data (80%, 70%, and 50%), we scrutinized the model's efficacy in distinguishing between different fine art categories. The metrics of accuracy, specificity, precision, recall, and F-score comprehensively assess the model's classification capabilities.

This systematic evaluation underscores the model's overall proficiency, offering nuanced insights into its effectiveness across different scenarios, which is crucial for understanding its robustness and applicability in real-world contexts.

Table 5 presents a comprehensive overview of the simulation results obtained using our proposed CNN and feature extraction method. Fine art images were classified using the K-fold method, with distinct configurations such as 10-fold, 5-fold, and 2-fold. This table provides a breakdown of various performance metrics, shedding light on the effectiveness of our approach. These metrics encompass vital indicators such as accuracy, specificity, precision, recall, and F-score. The detailed analysis across different K-fold configurations demonstrates the robustness and generalization capabilities of the proposed method in the context of fine art image classification.

For example, the accuracy of the proposed CNN and feature extraction method reached 93.34%, 93.24%, and 93.09% in 2-fold, 5-fold, and 10-fold cross-validations, respectively. These results highlight the robust performance of the model across the K-fold validation technique.

These visualizations are invaluable tools for evaluating the convergence, generalization, and overall effectiveness of the CNN model in classifying fine arts. They enable a thorough analysis of the model's learning dynamics, performance trends, and capability to accurately classify fine art images into these specified genres, objective types, techniques, and departments, contributing to the advancement of artificial intelligence in fine art recognition.

## 5.3 Comparison to other classifications

To comprehensively assess the effectiveness of our proposed method, we conducted a thorough comparative analysis against traditional machine-learning classification algorithms, namely Bayesian Network (BN), k-Nearest Neighbors (KNN), SVM, and RF [22–25]. The results, showcased in Table 6, highlight the performance of our proposed CNN and feature extraction method, employing both the holdout method with training data proportions set at 80% and 10-fold. This table meticulously presents crucial performance metrics, including accuracy and specificity, offering a holistic and detailed assessment of our model's

**Table 6. Performance comparison of proposed CNN and feature extraction method with traditional machine learning algorithms for fine art image classification.**

| Methods | Acc | Spe |
|---|---|---|
| Proposed Method (80% holdout) | 93.24% | 93.65% |
| Proposed Method (10-fold) | 93.34% | 93.70% |
| Convolutional neural network (CNN) [17] | 90.58% | 90.84% |
| Bayesian network (BN) [22] | 78.37% | 79.56% |
| k-nearest neighbors (KNN) [23] | 75.41% | 78.36% |
| Support vector machine (SVM) [24] | 83.27% | 88.64% |
| Random Forest (RF) [25] | 87.63% | 88.01% |

classification capabilities. For example, the accuracy results for our proposed method and the traditional algorithms—CNN, BN, KNN, SVM, and RF—are 93.24%, 93.34%, 90.58%, 78.37%, 75.41%, 83.27%, and 87.63%, respectively. Overall, our proposed CNN model outperforms all traditional classification models.

Fig 5 compares accuracy and specificity across different classification methods. Our proposed method, evaluated under an 80% holdout and a 10-fold cross-validation scheme, demonstrates superior performance to conventional machine learning algorithms such as CNN, BN, KNN, SVM, and RF. Specifically, the proposed methods achieve accuracy scores of 93.24% and 93.34%, respectively, surpassing the performance of the other methods. Moreover, the specificity scores of the proposed methods (93.65% and 93.70%) indicate their ability to identify true negatives correctly. Based on this figure, our proposed method exhibits promising results, indicating its effectiveness in the context of the studied classification task.

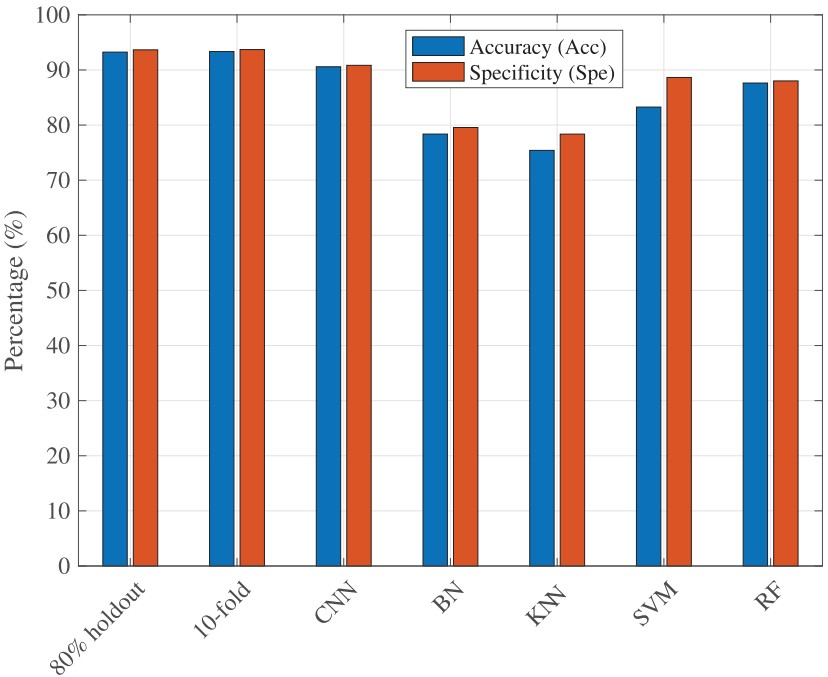

**Fig 5. Comparative analysis of classification methods: A focus on accuracy and specificity.**

## 6 Conclusion

This paper rigorously explores automated fine art recognition methodologies, culminating in introducing an innovative approach designed to enhance accuracy and efficiency significantly. Our method harnesses critical dataset attributes such as objective type, genre, material, technique, and department, demonstrating exceptional performance in classifying fine art pieces across diverse attributes. Our approach showcases remarkable performance by integrating advanced feature extraction techniques and a tailored CNN architecture, substantially advancing automated fine art recognition. Furthermore, this technological leap holds promise for numerous applications, including art history research, cultural heritage preservation, and digital art curation. It highlights its profound impact on the intersection of technology and art appreciation. To comprehensively capture these advancements, future research could explore additional dimensions such as temporal analysis of artistic styles and further integration of interdisciplinary perspectives in art and technology.

## Author Contributions

**Data curation:** Ruhua Chen.

**Formal analysis:** Ruhua Chen.

**Funding acquisition:** Ruhua Chen.

**Investigation:** Ruhua Chen.

**Methodology:** Mohammad Reza Ghavidel Aghdam.

**Project administration:** Mohammad Reza Ghavidel Aghdam.

**Resources:** Mohammad Khishe.

**Software:** Mohammad Reza Ghavidel Aghdam, Mohammad Khishe.

**Supervision:** Mohammad Khishe.

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
