## [Decision Letter · Decision Letter 0]

9 Jul 2024

PONE-D-24-17832Utilization of Artificial Intelligence for the Automated Recognition of Fine ArtsPLOS ONE

Dear Dr. Ghavidel Aghdam,

Thank you for submitting your manuscript to PLOS ONE. After careful consideration, we feel that it has merit but does not fully meet PLOS ONE’s publication criteria as it currently stands. Therefore, we invite you to submit a revised version of the manuscript that addresses the points raised during the review process.

We look forward to receiving your revised manuscript.

Kind regards,

Haiyan Li

Academic Editor

PLOS ONE

3. PLOS requires an ORCID iD for the corresponding author in Editorial Manager on papers submitted after December 6th, 2016. Please ensure that you have an ORCID iD and that it is validated in Editorial Manager. To do this, go to ‘Update my Information’ (in the upper left-hand corner of the main menu), and click on the Fetch/Validate link next to the ORCID field. This will take you to the ORCID site and allow you to create a new iD or authenticate a pre-existing iD in Editorial Manager. Please see the following video for instructions on linking an ORCID iD to your Editorial Manager account: https://www.youtube.com/watch?v=_xcclfuvtxQ.

Additional Editor Comments (if provided):

**Comments to the Author**

1. Is the manuscript technically sound, and do the data support the conclusions?

Reviewer #1: Yes

Reviewer #2: Partly

2. Has the statistical analysis been performed appropriately and rigorously? 

Reviewer #1: Yes

Reviewer #2: Yes

3. Have the authors made all data underlying the findings in their manuscript fully available?

Reviewer #1: Yes

Reviewer #2: Yes

4. Is the manuscript presented in an intelligible fashion and written in standard English?

Reviewer #1: Yes

Reviewer #2: Yes

5. Review Comments to the Author

Reviewer #1: The manuscript entitled Utilization of Artificial Intelligence for the Automated Recognition of Fine Arts in which the authors proposed a new AI-based approach in fine art recognition based on Convolutional Neural Networks

(CNNs). The manuscript is based on a good idea and can be accepted for publication but after some minor changes.

The authors should revise their manuscript according to the following comments.

Abstract

Should be restructured.

Introduction

Latest literature should be cited.

Methods

Fine

Results and discussion

Some grammatical mistakes are there which should be removed.

Conclusion should be revised.

Reviewer #2: This manuscript mainly applies machine learning algorithms to art recognition. Overall, the article is well written. The author proposes a CNN network that effectively improves the accuracy of classification, and trains the neural network using a comprehensive dataset from the online collection of the St. Adele Museum. At the same time, a comparison was made with machine learning algorithms such as BN, KNN, SVM, RF, etc. Compared with other algorithms, CNN networks have higher accuracy in art recognition and have certain academic value.

However, there are the following issues that need to be addressed:

(1) The innovation of the article should be highlighted more. CNN networks are not a completely new type of network in the field of machine learning. The author mentioned improvements to CNN networks in the article. Therefore, the comparison between traditional CNN networks and improved CNN networks in art recognition accuracy should be increased, not just with algorithms such as BN, KNN, SVM, etc.

(2) There are some issues with inconsistent image clarity in the manuscript, such as Figures 3 and 5. The clarity of the coordinate axis urgent legend in Figure 3 is too low.

(3) The conclusion of this manuscript is too simplistic and should be more comprehensive.

6. PLOS authors have the option to publish the peer review history of their article (what does this mean?). If published, this will include your full peer review and any attached files.

Reviewer #1: No

Reviewer #2: No

---

## [Author Response · Author response to Decision Letter 0]

17 Jul 2024

Reviewer 1 

Reviewer Point P 1.1 — Abstract: Should be restructured. Author response: 

Thank you for the reviewer’s valuable feedback. We have restructured the abstract accordingly. 

Authoraction: Hereistherestructuredabstract:

The recognition of fine art, traditionally dependent on human expertise, is undergoing a significant transformation with the integration of Artificial Intelligence (AI) and deep learning. This article introduces a novel AI-based approach for fine art recognition, utilizing Convolutional Neural Networks (CNNs) and advanced feature extraction techniques. Addressing the inherent challenges within this domain, we present a systematic methodology to enhance automated fine art recognition. By leveraging critical dataset characteristics such as objective type, genre, material, technique, and department, our method exhibits exceptional performance in classifying fine art pieces across diverse attributes. Our approach significantly improves accuracy and efficiency by integrating advanced feature extraction techniques with a customized CNN architecture. Experimental validation on a benchmark dataset highlights the efficacy of our method, indicating substantial contributions to the interdisciplinary field of fine art analysis. 

Reviewer Point P 1.2 — Introduction: Latest literature should be cited. 

Author response:

Thank you for your comment. I appreciate your feedback. I have now updated the introduction to include references to the latest literature:

In [16], the potential of established machine learning techniques to distinguish between authentic and imitation Pollocks is explored. A novel image ingestion method is developed, achieving 98.9% accuracy by decomposing images into multi-scaled tiles and utilizing transfer learning. Reference [17] explores fine-tuning CNNs for art-related image classification tasks, achieving state-of-the-art artist, genre, style, period, and national artistic context classification results. In [18], the focus was on advancing deep learning systems for automatic style classification by employing seven pre-trained EfficientNet models and custom architectures. The authors in [19] proposed a feature fusion method that combines various features used in painting captioning. The ablation study demonstrates that their feature fusion method enhances the model’s performance by an average of 9.8%. 

Please let me know if any further revisions are needed.

References:

16. Smith JH, Holt C, Smith NH, Taylor RP. Using machine learning to distinguish between authentic and imitation 

Jackson Pollock poured paintings: A tile-driven approach to computer vision. PloS One. 2024;19(6):e0302962.

17. HuW,LiX,LiC,LiR,JiangT,SunH,HuangX,GrzegorzekM,LiX. A state-of-the-art survey of artificial neural net- works for whole-slide image analysis: from popular convolutional neural networks to potential visual transformers. 

Computers in Biology and Medicine. 2023;161:107034.

18. Menai, Baya Lina. Recognizing the artistic style of fine art paintings with deep learning for an augmented 

reality application. PhD thesis, Universite Mohamed Khider (Biskra-Algerie), 2023.

19. Yue Lu, Chao Guo, Xingyuan Dai, Fei-Yue Wang. Generating emotion descriptions for fine art paintings via 

multiple painting representations. IEEE Intelligent Systems. 2023;38(3):31–40. 1 

Author action:

We updated the manuscript by adding the following:

1- In [16], the potential of established machine learning techniques to distinguish between authentic and imitation Pollocks is explored. A novel image ingestion method is developed, achieving 98.9% accuracy by decomposing images into multi-scaled tiles and utilizing transfer learning. Reference [17] explores fine-tuning CNNs for art-related image classification tasks, achieving state-of-the-art artist, genre, style, period, and national artistic context classification results. In [18], the focus was on advancing deep learning systems for automatic style classification by employing seven pre-trained EfficientNet models and custom architectures. The authors in [19] proposed a feature fusion method that combines various features used in painting captioning. The ablation study demonstrates that their feature fusion method enhances the model’s performance by an average of 9.8%. 

2- New References: 

16. Smith JH, Holt C, Smith NH, Taylor RP. Using machine learning to distinguish between authentic and imitation Jackson Pollock poured paintings: A tile-driven approach to computer vision. PloS One. 2024;19(6):e0302962. 

17. HuW,LiX,LiC,LiR,JiangT,SunH,HuangX,GrzegorzekM,LiX. A state-of-the-art survey of artificial neural net- works for whole-slide image analysis: from popular convolutional neural networks to potential visual transformers. Computers in Biology and Medicine. 2023;161:107034. 

18. Menai, Baya Lina. Recognizing the artistic style of fine art paintings with deep learning for an augmented reality application. PhD thesis, Universite Mohamed Khider (Biskra-Algerie), 2023. 

19. Yue Lu, Chao Guo, Xingyuan Dai, Fei-Yue Wang. Generating emotion descriptions for fine art paintings via multiple painting representations. IEEE Intelligent Systems. 2023;38(3):31–40. 

Reviewer Point P 1.3 — Results and discussion: Some grammatical mistakes are there which should be removed. 

Author response:

Thank you for your comment. We have reviewed and corrected any mistakes to ensure clarity and precision in our presentation of the results. 

Reviewer Point P 1.4 — Conclusion should be revised. Author response: 

Thank you for your comment. We have revised the Conclusion to improve its clarity. 

Author action:

We updated the conclusion:

This paper rigorously explores automated fine art recognition methodologies, culminating in the introduction of an innovative approach designed to enhance accuracy and efficiency significantly. Our method harnesses critical dataset attributes such as objective type, genre, material, technique, and department, demonstrating exceptional performance in classifying fine art pieces across diverse attributes. Our approach showcases remarkable performance by integrat- ing advanced feature extraction techniques and a tailored CNN architecture, substantially advancing automated fine art recognition. Furthermore, this technological leap holds promise for numerous applications, including art history research, cultural heritage preservation, and digital art curation. It highlights its profound impact on the intersection of technology and art appreciation. To comprehensively capture these advancements, future research could explore addi- tional dimensions such as temporal analysis of artistic styles and further integration of interdisciplinary perspectives in art and technology. 

Reviewer 2 

Reviewer Point P 2.1 — The innovation of the article should be highlighted more. CNN networks are not a com- pletely new type of network in the field of machine learning. The author mentioned improvements to CNN networks in the article. Therefore, the comparison between traditional CNN networks and improved CNN networks in art recognition accuracy should be increased, not just with algorithms such as BN, KNN, SVM, etc. 

Author response:

Thank you for your valuable and constructive feedback. We appreciate your suggestion to highlight the innovation of our approach and provide a more comprehensive comparison with traditional CNN networks and improved versions in art recognition accuracy. 

Author action:

We updated the manuscript by adding the following:

1- We have emphasized the innovation of our approach, particularly focusing on enhancements made to CNN 

architectures specific to fine art recognition.

2- We have conducted additional experiments to compare the performance of traditional CNN networks with our 

improved CNN networks in art recognition accuracy and specificity. These results are presented in Table 6 and Figure 5 for direct comparison and evaluation. 

3- ” For example, the accuracy results for our proposed method and the traditional algorithms—CNN, BN, KNN, SVM, and RF—are 93.24%, 93.34%, 90.58%, 78.37%, 75.41%, 83.27%, and 87.63%, respectively. Overall, our proposed CNN model outperforms all traditional classification models.” is added. 

This action plan addresses the reviewer’s suggestion by highlighting the innovation of your approach and providing a comparative analysis between traditional and improved CNN networks in the context of art recognition accuracy. 

Reviewer Point P 2.2 — There are some issues with inconsistent image clarity in the manuscript, such as Figures 3 and 5. The clarity of the coordinate axis urgent legend in Figure 3 is too low. 

Author response:

We appreciate the reviewer’s observation and feedback. Ensuring clear and understandable visual presentation is crucial to communicate our findings effectively. We have taken steps to improve the quality of these figures. 

Author action:

We updated the manuscript by adding the following:

1- Enhanced the clarity of Figures 3 and 5 by improving the resolution of the coordinate axis and legends, ensuring better readability.

2- Revised the legends in Figure 3 to ensure they are clear and legible, addressing the specific concerns raised by the reviewer. 

Reviewer Point P 2.3 — The conclusion of this manuscript is too simplistic and should be more comprehensive. Author response: 

We appreciate the reviewer’s observation and feedback. We have revised the Conclusion to improve its clarity. 

Author action:

We updated the conclusion:

This paper rigorously explores automated fine art recognition methodologies, culminating in the introduction of an innovative approach designed to enhance accuracy and efficiency significantly. Our method harnesses critical dataset attributes such as objective type, genre, material, technique, and department, demonstrating exceptional performance in classifying fine art pieces across diverse attributes. Our approach showcases remarkable performance by integrat- ing advanced feature extraction techniques and a tailored CNN architecture, substantially advancing automated fine art recognition. Furthermore, this technological leap holds promise for numerous applications, including art history research, cultural heritage preservation, and digital art curation. It highlights its profound impact on the intersection of technology and art appreciation. To comprehensively capture these advancements, future research could explore addi- tional dimensions such as temporal analysis of artistic styles and further integration of interdisciplinary perspectives in art and technology.

---

## [Decision Letter · Decision Letter 1]

14 Oct 2024

Utilization of Artificial Intelligence for the Automated Recognition of Fine Arts

PONE-D-24-17832R1

Dear Dr. AGHDAM,

We’re pleased to inform you that your manuscript has been judged scientifically suitable for publication and will be formally accepted for publication once it meets all outstanding technical requirements.

Kind regards,

Delanyo Kwame Bensah Kulevome, Ph.D.

Guest Editor

PLOS ONE

Additional Editor Comments (optional):

Reviewers' comments:

Reviewer's Responses to Questions

**Comments to the Author**

1. If the authors have adequately addressed your comments raised in a previous round of review and you feel that this manuscript is now acceptable for publication, you may indicate that here to bypass the “Comments to the Author” section, enter your conflict of interest statement in the “Confidential to Editor” section, and submit your "Accept" recommendation.

Reviewer #1: All comments have been addressed

Reviewer #2: All comments have been addressed

2. Is the manuscript technically sound, and do the data support the conclusions?

Reviewer #1: Yes

Reviewer #2: Yes

3. Has the statistical analysis been performed appropriately and rigorously? 

Reviewer #1: (No Response)

Reviewer #2: Yes

4. Have the authors made all data underlying the findings in their manuscript fully available?

Reviewer #1: Yes

Reviewer #2: Yes

5. Is the manuscript presented in an intelligible fashion and written in standard English?

Reviewer #1: Yes

Reviewer #2: Yes

6. Review Comments to the Author

Reviewer #1: The authors have extensively revised the manuscript and have answered the comments of all reviewers. The manuscript can be accepted for publication now.

Reviewer #2: The author carefully revised the manuscript and addressed the issues I raised. Therefore, this article can now be accepted.

7. PLOS authors have the option to publish the peer review history of their article (what does this mean?). If published, this will include your full peer review and any attached files.

Reviewer #1: No

Reviewer #2: No

---

## [Editor Report · Acceptance letter]

18 Oct 2024

PONE-D-24-17832R1 

PLOS ONE

Dear Dr. Ghavidel Aghdam, 

I'm pleased to inform you that your manuscript has been deemed suitable for publication in PLOS ONE. Congratulations! Your manuscript is now being handed over to our production team.

Kind regards, 

on behalf of

Dr. Delanyo Kwame Bensah Kulevome 

Guest Editor

PLOS ONE